# Combining Value-Focused Thinking and PROMETHEE Techniques for Selecting a Portfolio of Distributed Energy Generation Projects in the Brazilian Electricity Sector

Mirian Bortoluzzi [1,2,*] , Marcelo Furlan [1] , Simone Geitenes Colombo [1,2], Tatiele Martins Amaral [1], Celso Correia de Souza [2,3], José Francisco dos Reis Neto [2,3] and Josimar Fernandes de França [3]

1 Department of Management Engineering, Universidade Federal do Mato Grosso do Sul (UFMS), Campus de Nova Andradina, Nova Andradina 79750-000, MS, Brazil; marcelo.furlan@ufms.br (M.F.); simone.g@ufms.br (S.G.C.); tatiele.martins@ufms.br (T.M.A.)
2 Programa de Pós-Graduação em Meio Ambiente e Desenvolvimento Regional, Universidade Anhanguera Uniderp, Campo Grande 79070-900, MS, Brazil; csouza939@gmail.com (C.C.d.S.); jose.rneto@educadores.net.br (J.F.d.R.N.)
3 Programa de Pós-Graduação em Produção e Gestão Ambiental, Universidade Anhanguera Uniderp, Campo Grande 79070-900, MS, Brazil; josimargtl@gmail.com
* Correspondence: mirian_bortoluzzi@ufms.br

**Abstract:** This article aims to propose a multi-criteria model to support decision-making from a portfolio in selecting technologies for Distributed Generation of Energy (DGE) projects based on the characteristics of the geographic space in Brazil. The decision model involves using multi-criteria to support the evaluation, prioritization, and selection of projects under a multistage decision-making process that fits into a strategic management cycle within the energy sector of Mato Grosso do Sul (Brazil). The over-classification techniques *Preference Ranking Organization Technique for Enrichment Evaluations (PROMETHEE) II* and *V* were applied under the *Value-Focused Thinking (VFT)* approach, reflecting the decision-maker or manager preferences among several conflicting criteria in the investment context of sustainable distributed energy generation projects. Based on real data, a numerical application is employed to view the steps of this decision model and illustrate the adequacy and effectiveness in practical issues of portfolio management.

**Keywords:** sustainable distributed generation; clean energy; portfolio selection; MCDM/A; PROMETHEE

## 1. Introduction

Energy is an essential element for human survival, and demand is growing every day [1]. Managers have experienced strong stakeholder pressures in their routines, mainly in their decisions focusing on the sustainable performance of organizations, as an evaluation, prioritization, and appropriate selection of resources to increase their competitive potential [2]. Project portfolio selection problems, meanwhile, become complex, as these involve conflicting criteria most of the time, and uncertainties both in the results generated by resource allocation policies and the strategic objectives defined by the company [3].

In the electrical sector, from the moment a company plans to invest in the selection of energy projects to compose its portfolio, several issues are directly associated with the technologies, innovation, and sustainability, as well as seeking to direct their efforts towards the development of new technological routes and investment options in the energy field [4]. If the linkage between these activities and their consequences are recognized, the need arises to propose new forms of planning, designing, and investing and, in doing so, to seek technically and economically feasible alternatives that encourage rationalization [5].

From this perspective, the importance of the decision-making process in the context of sustainability in the portfolio selections of electric energy generation projects can be

seen [6]. This requires structuring objectives and relating them to the implementation of sustainability initiatives based on the key stakeholders' values [7]. A manager often concentrates his efforts first on the alternatives, and, only after that, does he define his objectives or criteria that will then be evaluated [8]. The focus on the alternatives is a reactive and limited way to think about decision-making [9]. The time and effort spent on decision-making are used when thought is being given to values [10]. Therefore, the idea is to generate viable alternatives in accordance with the values of the managers involved, to help solve the decision-making problems being studied [11].

In recent years, studies have been conducted to contribute to sustainability in the portfolio selection context of energy projects [12,13]. For example, the application of Value-Focused Thinking (VTF) has been a useful tool in the proposition of decision models for decision-making about energy performance [14], energy planning [15], energy improvement [16], and, among other applications, contributing important insights for decision-making [17].

However, there is no known application in the portfolio selection of a DGE project's environmental, economic, and social constraints in developing geographical regions. This study focuses on the complement of hydroelectric plants, selecting DGE projects, and identifying their role in the electro-energy dispatch of the State of Mato Grosso do Sul. It also considers different regions of the state in terms of the potential generation of clean and accessible energy for the state's most remote regions. This is a planning problem, which requires a multi-criteria decision and involves a large number of alternatives. Multiple criteria, stakeholders in the decision-making process, and the application of the MCDM/A conform technique are explained by Jajac et al. [18].

Therefore, this paper aims to propose a multi-criteria model to support decision-making with regards to a portfolio on the selection of technologies for DGE projects, based on the characteristics of geographic space in Brazil. To achieve this goal, the proposed model is based on the combination of VFT and PROMETHEE techniques, whose application allows us to take in sustainability characteristics where the best options for power generation will be chosen. This paper advances knowledge because it considers the structuring—within the decision-making process—of feasible alternatives for portfolio selection in the sustainable electric energy distributed generation context.

The object of study is the State of Mato Grosso do Sul, located in Brazil. This State has unique characteristics that impose serious restrictions on the elaboration and implementation of electricity distribution projects. Although the Brazilian electric matrix has as its main source hydroelectric power plants (i.e., 74% of the electric matrix) [19], the reservoirs that supply the plants in the midwest region already show a decrease in volume and, consequently, a decrease in the capacity to generate energy [20]. Another important feature is that the Pantanal biome is mostly in Mato Grosso do Sul [21], considered a sanctuary of endangered species [22]. Thus, the choices of electricity distribution projects must consider biodiversity, water management, and climate change issues so that the population has energy from renewable sources and the country will achieve its Sustainable Development Goals.

Thereby, this paper seeks to better understand this problem and do this using the VFT technique combined with the PROMETHEE technique. The Value-Focused Thinking (VFT) method thereby seems advantageous compared to other traditional techniques applied in decision-making in the sustainability context [23]. The PROMETHEE ("Preference Ranking Organization Technique for Enrichment Evaluations"), according to Debbarma et al. [24], is used for decision making to rank a finite set of alternatives based on selected criteria/indicators that, sometimes, are conflicting. Combining these techniques allows for attenuating both the restrictions and values existing in the management decision and providing the best alternatives for the analyzed context.

Thus, it is expected that, as a result of reading this study, project managers who work in energy companies may have a better problem understanding. Thereby, it is hoped that, by providing a model for the objectives structuring, this study will be used as a

starting point for implementing sustainable in the portfolio selection of energy distributed generation projects.

This study aims to give a new viewpoint to both project managers of energy companies and society and how best to define measures for the financial investment in sustainability in electric energy distributed generation projects. This is one of the main factors considered today in deciding whether to implement sustainable energy projects. Nevertheless, as will be shown, as well as the projects' investment cost, resource allocation should also be identified and considered before making such decisions.

This article is organized as follows: Section 1 presents the introduction; Section 2 offers a literature review of the application context of MCDM/A and VFT methods; Section 3 presents the research technique and describes how the decision objectives to implement sustainability in the portfolio selection of energy distributed generation projects were structured; Section 4 presents a numerical application; Section 5 discusses the results obtained; Section 6 presents the conclusions of this research, making suggestions for future studies.

## 2. Literature Review

Several studies address how better to evaluate the alternatives when compiling a portfolio [25]. Among these studies, some identify various methodologies with the support of Multicriteria Decision-Making/-Aiding (MCDM/A) techniques, especially those used in areas such as Research and Development (R&D) projects [26], infrastructure asset management [27], capital budgeting for health care [28], power plant investment projects [29], and construction systems [30].

As regards applications, the literature supports MCDM/A, especially in the electric energy sector. For example, Aragónes–Béltran et al. [31] present an approach to select investment projects in photovoltaic solar energy plants. They apply the Analytic Hierarchy Process (AHP) and the Analytic Network Process (ANP) to help the management board of a Spanish solar energy investment company decide whether to invest in a particular solar thermal power plant project and, if so, to determine the priority order of projects in the company's portfolio.

Lourenço et al. [32] presented a Portfolio Robustness Evaluation (PROBE) model, developed by an electricity distribution company to select the best project portfolios, subject to budget constraints for different project types. Jano–Ito and Crawford–Brown [33] presented a framework for the combination of decision-making approaches that have been considered separately using Mean-Variance Theory (MVT) and Multi-Attribute Utility (MAUT) portfolios, and this framework is applied to the Mexican electric sector considering the attitudes toward risk and the preferences of the main firms.

Büyükozkan and Karabulut [34] proposed a multi-criteria decision model with a sustainability perspective to improve the selection of a highly detailed energy project. Martins et al. [35] proposed a model for selecting a project portfolio in an electricity company in Brazil, a compensatory-based approach using an additive value function. Wu et al. [2] proposed a multi-criteria framework for selecting project portfolios for power generation, considering uncertainty and project interaction under different strategic scenarios.Therefore, the IT2FAHP technique, is utilized to select energy projects.

The decision models presented above aim to support decisions in energy planning, focusing on evaluating exchanges between multiple performance criteria. However, for the portfolio selection of energy projects, the above decision models allow for evaluating compensation between multiple criteria simultaneously, to obtain a solution that represents a decision-maker's preference. On the other hand, the great difficulty presented by such methods is the elicitation of the scale constants and the obtaining of partial information provided by the decision-maker, which requires a greater cognitive effort to answer the trade-off [36].

Hernandez–Perdomo et al. [37] evaluated data involving economic, management, and social perspectives for project selection in state-owned energy companies. The

PROMETHEE II outranking method was applied to rank projects, among which one or more would be selected and constructed. We also supposed that the company had a budget constraint, and the selected projects would be those that satisfy that restriction, which would assist in the decision-making process.

Guler et al. [38] suggested using the Data Envelopment Analysis (DEA) model and PROMETHEE technique to evaluate sustainable energy alternative performance for 36 countries of the Organization for Economic Cooperation and Development.

Brans and Vincke [39] have already confirmed in their study that the PROMETHEE technique is quite simple in its conception and application compared to other over-classification methods. The weights of the criteria are associated with the degree of importance assigned by the decision-maker. In this case, this occurs due to the use of non-compensatory rationality. The use of this method does not require exchange procedures to quantify them as with compensatory methods. This makes cognitive understanding of the weight allocation process simpler for decision-making stakeholders.

Traditional techniques generate criteria after analyzing decisions by studying alternative methodologies known as Alternative-Focused Thinking (AFT) [8]. These techniques restrict the focus of attention to the available alternatives, which prevents criteria that express the importance of values to decision-makers from your preferences [8].

In Value-Focused Thinking (VFT), the criteria appear when the set of values for the decision context is defined. Such values can include the purpose of the desire to identify what is important [8]. Thus, the alternatives are defined to meet the decision-maker's objective. This allows for the emergence of new ideas that contribute to rationalizing the use of technologies in electric energy distributed generation to positively impact environmental sustainability.

However, it is observed that the sustainability implementation values in electric energy distributed generation portfolio selection need to be understood and structured. To the authors' best knowledge, no studies have been conducted that structured this question by incorporating the views that different actors have of the problem, and no study considered the different relevant aspects jointly to implement and maintain sustainability. To diminish this gap in the literature, this paper used the VFT technique application to structure objectives and create project actions in a structured way.

With structured objectives and known alternatives, decision-making techniques can be applied to improve the quality of decisions, in the sustainability context in electric energy distributed generation. Furthermore, we apply the PROMETHEE II and V techniques, subsequently described, to rank the projects and select the optimal set according to the resource availability.

Thus inspired, the PROMETHEE technique and the VFT methodology are used, considering regional sustainability characteristics and where the best options for power generation will be chosen in this study.

## 3. Material and Method

### 3.1. Value-Focused Thinking (VFT) Techniques

The VFT technique proposed by Keeney [8] is based on values that aim to discover unknown strategic objectives in the most diverse managerial processes [8,40]. The VFT technique application is composed of four steps:

Step 1. Stakeholders' identification: the stakeholders are important actors in the decision-making, directly linked to the organization [36]. In project portfolio management, the stakeholders must be partners and perform an essential role in evaluating the projects that will compose the portfolio.

Step 2. Stakeholders' values identification: values are fundamental principles to the decision-making process [8,40]. Therefore, adopting a structured approach to understanding the stakeholders' main values allows for the identification of better decision situations. Thus, in the decision domains about the portfolio selection of distributed electric energy

generation, under a sustainable perspective, this can allow for identifying the technologies used that are related to sustainability concepts.

Step 3. Converting values into goals: three characteristics are particular to a goal: a decision context, object, and preferred direction [8]. This essentially means explaining the goal within its context based on the problematic nature and determining exactly what the stakeholder is trying to achieve. This further underscores the importance of interacting with the stakeholders to understand their vision and values better. The objectives are classified into two types: (i) fundamental objectives—those that configure the purposes of decision-makers' values in a decision-specific context; (ii) means objectives—the techniques of achieving these purposes [8], which are independent contexts. This implies that natural consideration, or the particular purpose, will determine how the fundamental objectives and means will be formed.

Step 4. Determine the relation between objectives: to accomplish this step, one must ask why each objective is important, helping the decision-maker to distinguish between the fundamental objectives and the means objectives [8,40], which result in the means-end objective network. Thereby, the VFT technique helps the decision-makers to see the alignment of strategic portfolio objectives.

From this perspective, the VFT technique presents a proactive pattern for directing the decision-maker's decision and behavior, seeking decision opportunities under a specific context, in this case, the portfolio selection of DGE projects. This approach is designed, then, to identify desirable decision opportunities and create alternatives through considerable effort in making values explicit, allowing the decision-maker greater control over the situation he faces. In the context of this study, the VFT application is justified because managers have been looking for indicators that support the Sustainable Development Goals in their decisions [41].

Having defined the alternatives (projects) to be considered in the problem under study, an MCDM/A technique is chosen to produce a suitable recommendation for the DM, given the problem characteristics, which drives the present research to highlight the study of over-classification techniques, specifically the PROMETHEE II and PROMETHEE V techniques, presented in the following subsections.

### 3.2. PROMETHEE II

The PROMETHEE technique, according to Brans and Vincke [39], is an overcoming technique that uses pairwise comparisons between alternatives to building the overcoming relation and exploits it to support the decision process. Instead of pointing at "correct" decisions, the PROMETHEE technique helps decision-makers find the alternative that best fits their objective and problem understanding.

To illustrate the concept mentioned above, the following notations for the projects and criteria for the next step are used. Let A = $\{a_1, \ldots, a_j, \ldots, a_n\}$ be a set of alternatives evaluated on a set of $q$ criteria F = $\{f_1, \ldots, f_t, \ldots, f_q\}$ and let $f_k(a)$ be the performance of alternative $a \in$ A, evaluated in terms of the decision criterion $f_k$, $k = 1, \ldots, q$. Without loss of generality, it is assumed that these criteria should be maximized. These data can be written in a table n $\times$ q containing the evaluations; in this way, each row corresponds to an alternative, and each column corresponds to a criterion (Table 1).

Given two alternatives, $a_i$ and $a_j$, the main PROMETHEE steps can be described in the following steps:

Step 1: specification of the preference structure. For criterion $f_k$, this structure translates the difference $d_k$, between the evaluation of two alternatives, Equation (1).

$$d_k(a_i, a_j) = f_k(a_i) - f_k(a_j); \forall a_i, a_j \in A, \forall k = 1, \ldots, q \tag{1}$$

The notion of preference function, denoted by $\pi_k(a_i, a_j)$, is introduced to translate the degree of difference within a preference criterion, as per Equation (2).

$$\pi_k(a_i, a_j) = P_k[d_k(a_i, a_j)] \tag{2}$$

where $P_k: \mathcal{R} \to [0,1]$ is a non-decreasing positive preference function such that $P_k(0) = 0$.

**Table 1.** Decision-Matrix.

|       | $f_1(.)$  | $f_2(.)$  | ...   | $f_j(.)$  | ...   | $f_q(.)$  |
| ----- | --------- | --------- | ----- | --------- | ----- | --------- |
| $a_1$ | $f_1(a_1)$ | $f_2(a_1)$ | ... | $f_j(a_1)$ | ... | $f_q(a_1)$ |
| $a_2$ | $f_1(a_2)$ | $f_2(a_2)$ | ... | $f_j(a_2)$ | ... | $f_q(a_2)$ |
| ...   | ...       | ...       | ...   | ...       | ...   | ...       |
| $a_i$ | $f_1(a_i)$ | $f_2(a_i)$ | ... | $f_j(a_i)$ | ... | $f_q(a_i)$ |
| ...   | ...       | ...       | ...   | ...       | ...   | ...       |
| $a_n$ | $f_1(a_n)$ | $f_2(a_n)$ | ... | $f_j(a_n)$ | ... | $f_q(a_n)$ |

The PROMETHEE uses six types of generalized criteria, namely: type 1 (usual), type 2 (quasi-criterion), type 3 (linear preference criterion), type 4 (level criterion), type 5 (linear preference criterion with an indifference zone), and type 6 (Gaussian); in this study type 1 was used. According to Brans and Vincke [39], a preference function is the unicriteria linear preference function, Equation (3).

$$P_k(x) = \left\langle \begin{array}{cc} 0, & se\ x\ \leq r_k \\ \frac{x-r_k}{p_k-r_k} & se\ r_k < x\ \leq p_k \\ 1 & se\ x > p_k \end{array} \right. \tag{3}$$

where $q_k$ and $p_k$ are the indifference threshold $r$ and the preference threshold $p$, respectively. The meanings of these parameters are as follows: when the difference between them is smaller than the indifference threshold, it is considered insignificant for deciding. However, the corresponding degree of unicriterion preference is equal to zero; otherwise, it is significant. In this case, the degree of unicriterion preference is equal to 1 (maximum value). When the preference is between two thresholds, an approximate value of the degree of preference can be obtained by linear interpolation.

Step 2: The comprehensive preference index for the preference of $a_i$ over $a_j$, for all pairs of alternatives $a_i, a_j \in A$ is given by Equation (4).

$$\pi_k(a_i, a_j) = \sum_{k=1}^{q} \left[ w_k P_k(a_i, a_j) \right] \tag{4}$$

where *wk* are positive weights associated with the different criteria by the DM, with $\sum_{k=1}^{q} w_k = 1$. The consequences are Equations (5) and (6).

$$\pi_k(a_i, a_j) \geq 0 \tag{5}$$

$$\pi_k(a_i, a_j) + \pi_k(a_j, i) \leq 1 \tag{6}$$

Step 3: The positive flow scores $\Phi^+$ and negative flow scores $\Phi^-$ exceeded are defined according to Equations (7) and (8).

$$\Phi^+(a_j) = \frac{1}{n-1} \sum_{a_j \in A} \pi(a_i, a_j) \tag{7}$$

$$\Phi^-(a_j) = \frac{1}{n-1} \sum_{a_i \in A} \pi(a_j,\ a_i) \tag{8}$$

Step 4: In the PROMETHEE II technique, the complete ranking of alternatives using net flows $\Phi^+(a_j)$ is obtained by Equation (9):

$$\Phi(a_j) = \Phi^+(a_j) - \Phi^-(a_j) \tag{9}$$

The higher the flow, the better the ranking. In this approach, only the relations *p* of preference and indifference *r* are possible when comparing each pair of alternatives.

Thus, at this step, one has the main contribution to the decision model. In other words, this is where the logical structure to calculate the net flow is produced, that is, the objective function used to build the PROMETHEE V approach (described in the following section). Then, it is possible to process and provide information about a selection decision process in this portfolio context.

### 3.3. PROMETHEE V

At this point, PROMETHEE V is calculated using net flows $\Phi(a_j)$, obtained with the PROMETHEE II technique. The solution to PROMETHEE V developed by Brans and Mareschal [42] is obtained by Equations (10)–(12).

$$Maximize\ V(p_r)\ \sum_{j=1}^{n} \Phi(a_j)\, x_j \tag{10}$$

$$S.\,t.: \sum_{j=1}^{n} b_j x_j \leq B \tag{11}$$

$$x_j \in \{0,1\}, \forall_j\, j = 1, \ldots, n \tag{12}$$

where *xj* is a binary variable that indicates whether or not item $x_j$ is included in the portfolio; therefore, $x_j = 1$, if it is included, and $x_j = 0$, if it is not. $\Phi(a_j)$ is the item value obtained with the PROMETHEE II technique; *B* and $b_j$ are connected to the constraints, where *B* is considered the budgeted value available to finance the project costs, and $b_j$ is the cost to develop project *j*. This is a constraint to the portfolio problem, being the budget constraint on the investment in projects.

According to the budget constraint, each project alternative consumes a certain amount, to be implemented in the project portfolio. The total amount to be spent by all projects in a portfolio must not exceed the limit defined in the budget, generating a minor or equal constraint to the problem.

### 3.4. Decision Framework of Portfolio Selection of DGE Project Portfolio

The Decision Framework to Project Portfolio Selection (DFPPS) provides a formal approach to assist in developing objectives and definitions of the DGE projects that represent the values of the various stakeholders in the decision-making process on which the best DGE projects are.

The DFPPS involves a series of interactive steps to help identify the stakeholders' desired goals and the measures of the DGE projects that are directly associated with these goals. Stakeholder identification is necessary to better consider the essential values to the perception of portfolio success. The definition of the actors in the decision-making process follows the methodology proposed by De Almeida et al. [36]. Regarding the definition of decision-making and stakeholders, the energy project manager was assigned the role of decision-maker. At the same time, the operational leaders of each of the sectors and the analyst of techniques and processes were characterized as experts. In addition, it is important to evaluate the presence of stakeholders (such as shareholders and government representatives) whose influence must be carefully managed to prioritize the organization's interests.

The activities for eliciting and structuring represent the formative stages of criteria design and project definition. In order to achieve its objective, the DFPPS is based on the

methodological principles already established in the literature. The VFT and PROMETHEE were chosen partly because of their high cohesiveness, i.e., VFT focuses on a structured approach to obtain objectives. In contrast, the PROMETHEE II and V techniques were chosen due to their non-compensatory rationale and the visual software (http://www.promethee-gaia.net/vpa.html accessed on 20 January 2021).

The proposed decision DFPPS consists of three main stages, including the stage of obtaining preliminary information through the VFT technique, such as the definition of the criteria sets and the set of proposed projects; the second stage consists of building a decision matrix using PROMETHEE II, in which the decision-maker will evaluate the projects on the criteria and the definition of their weights. The last stage consists of applying PROMETHEE V to obtain the optimal combination of projects and, thus, forming a portfolio of DGE sustainable projects. The combination of techniques supports and facilitates stakeholder participation, thus improving the performance of organizational portfolios (Figure 1).

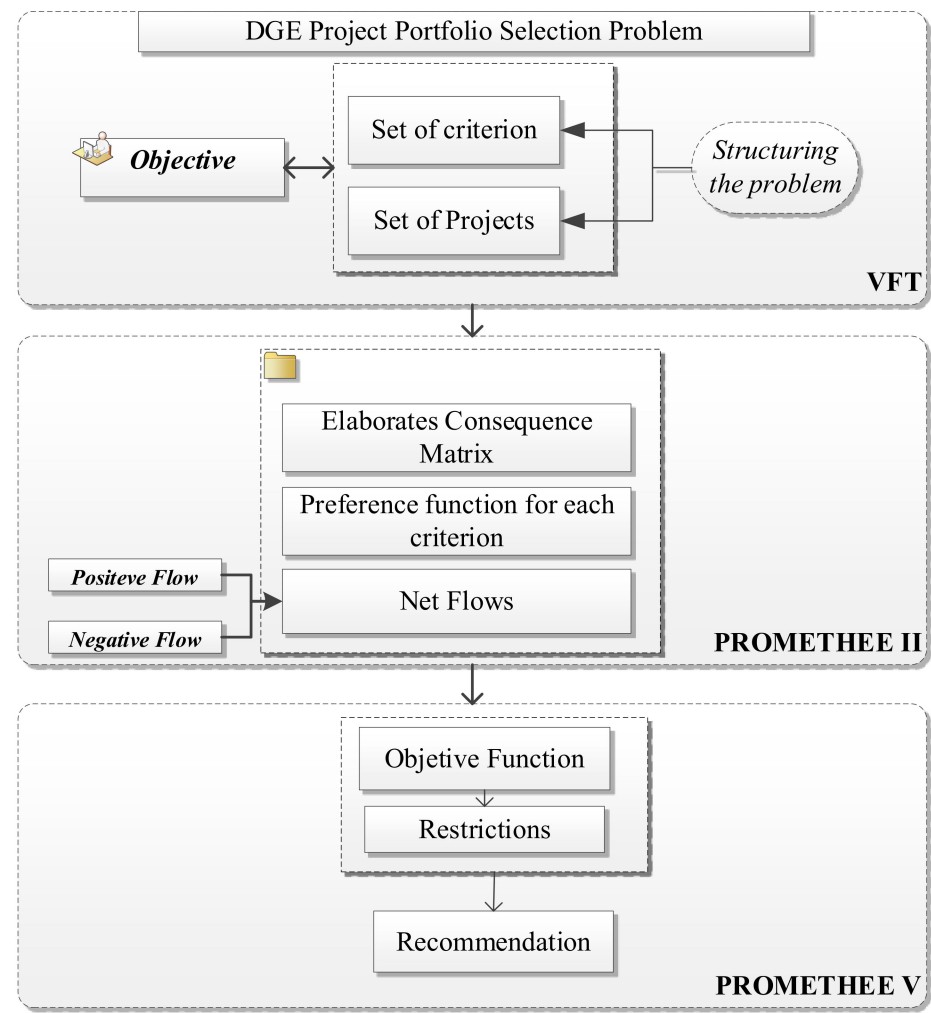

**Figure 1.** Decision DFPPS for DGE project portfolio selection.

The proposed DFPPS consists of three stages (Figure 1), illustrated in the next section, through a case in an electric sector context, as well as the development of the application of this model. Figure 1 illustrates the proposed model in this study, which has three main phases, including identifying the problem and structuring the problem by the VFT; the project net flow obtained by the PROMETHEE II technique; the PROMETHEE V technique to select the project. Each phase in the suggested proposed model and their corresponding steps are described in detail in Section 4.

## 4. Results and Discussion

### 4.1. Problem Characterization

The Brazilian electric energy model is based on generating large hydroelectric plants, with transmission through high-voltage lines and subsequent low-voltage distribution to end consumers [43]. This model requires high investment in large plants because many of them are located far from consumption centers, especially the plants included in the sectorial planning, which requires high investment to reach consumers of the energy generated [44].

The development of electric energy distributed generation can change this paradigm by generating energy close to the units of consumers [2]. In the world, the expansion of distributed generation (DG) had as a motivating factor the reduction of greenhouse gas emissions (GHG) [43]. Many countries are using distributed generation from renewable sources, mainly solar photovoltaic, to replace fossil fuel generation to reduce carbon dioxide emission [45].

In Brazil, the Minas and Energy Ministry (MME) launched the Program for Development of Electric Energy Distribute Generation (ProGD) to broaden and deepen the actions to stimulate the energy generation by consumers themselves, based on renewable energy sources, especially solar, photovoltaic, wind, and biomass, to support and clarify energy companies investing in energy projects, which DG aggregates, both in terms of the National Agency of Electric Energy (ANEEL) [46–48].

The state of Mato Grosso do Sul is the 6th state in the country in terms of territorial extent, with 357,145,534 km$^2$. This corresponds to 4.19% of the total area of Brazil (8,515,767.049 km$^2$) and 22.23% of the area of the central region-West. The area's estimated population in 2021 is 2,839,188 inhabitants, putting the state 21st in Brazil in terms of population. Its capital and largest city is Campo Grande, and other important municipalities are Dourados, Três Lagoas, Corumbá, Ponta Porã, Aquidauana, Nova Andradina, and Naviraí. The intrinsic character of the micro- and meso-regional divisions of Mato Grosso do Sul refer to a set of economic, social, and political determinations that concern the entire organization of space in the state territory. The aims are to assist in the development of public policies, planning, and the subsidizing of regionalized and local studies, according to the Brazilian Institute of Geography and Statistics (IBGE) [49]. Mato Grosso do Sul has in agriculture one of its greatest economic forces; the region stands out for its very diversified production in agribusiness, being a large supplier of various products such as sugarcane, corn, cattle, soybeans, and cellulose [49]. In the energy sector, in 2018, the state's electricity consumption was 5,788,274 MWh, and its consumers numbered 1,089,139, according to the State Secretaria for Environment and Economic Development, Production, and Family Agriculture (SEMAGRO) [50]. The State of Mato Grosso do Sul has 2 hydroelectric power plants, 9 small hydroelectric power plants, 1 solar photovoltaic generating plant, 21 thermoelectric power plants based on natural gas in the fossil class, and 28 thermoelectric power plants using sugarcane bagasse as fuel in the biomass class.

The mass insertion of DG can bring benefits beyond the electricity sector, such as job generation and economic and sustainable development, in a moment in which the country is going through difficulties, both in the economic area and in employment level.

From this perspective, the aim is to subsidize decision-makers in government, the private sector, and other societal sectors interested in the Brazilian electric energy question, under an investment analysis in DGE projects in a sustainable way. In this case, the study object was the State of Mato Grosso do Sul (MS), a potential investment area because of its abundant natural resources, such as wind, sun, and biomass, mainly those that come from organic matter [20].

Thus, this study seeks to evaluate the alternatives related to DGE potential generation projects in selected locations in the State, from a sustainable, environmental, technological, social, and economic perspective, in such a way that, in order to have a sustainable hybrid portfolio, some projects should be selected to compose it.

### 4.2. Portfolio Selection of Energy Distributed Generation Projects

In the first stage, the actors of the decision-making process, such as managers and experts, are defined. As the model is intended for multi-criteria problems, in which multiple objectives are to be achieved, the establishment of such fundamental objectives, middle-end objectives networks, a set of criteria related to the objectives, and potential projects are relevant steps to build the model as a VFT application result.

At this stage, the strategic objective of the research was defined, which consisted of proposing potential projects that increase the electric energy distributed generation with renewable energy sources in rural and industrial facilities and have a positive impact on the technological, social, economic, and environmental dimensions.

Next, a list of key objectives was defined, structured into three development strategies: sustainable (environmental), socioeconomic, and technological. Thus, the fundamental objectives associated with this research purpose were structured as presented in Table 2.

**Table 2.** Hierarchy of key objectives of development strategies.

| **1.** **Promoting Sustainable Development** |
|---|
| 1.1 Harnessing the potential of Mato Grosso do Sul |
| 1.2 Generating energy from clean and renewable sources |
| 1.3 Reducing energy waste |
| 1.4 Reducing greenhouse gas emission |
| 1.5 Lower the impacts of non-renewable source projects |
| **2.** **Promoting Socioeconomic Development** |
| 2.1 Generating employment and income |
| 2.2 Reducing energy losses |
| 2.3 Universalizing access to energy |
| 2.4 Reducing energy bills |
| 2.5 Strengthening energy and electric security |
| **3.** **Promoting Technological Development** |
| 3.1 Promoting attraction of international investment and renewable technology |
| 3.2 Encouraging the equipment industries with a focus on technology and sustainability |
| 3.3 Spread the use of clean and efficient technologies |
| 3.4 Reducing the institutional and market barriers for new technologies |

Once the fundamental objectives that provide the context for this decision are composed, the middle objectives can define the potential projects that achieve the fundamental objectives. Thus, the network structure of middle-end objectives was defined, as shown in Figure 2.

The environmental, socioeconomic, and technological dimensions were considered three main categories in this study [41]. However, resource allocation, use of a freely available resource or renewable sources of energy, and the awareness of consumers, farmers, industries, and trades play key elements in a sustainable hybrid portfolio and are considered middle objectives.

Based on the dimensions mentioned above and the defined fundamental objectives, the model proposed in this research comprises a set of seven evaluation criteria (Table 3). In the literature, there are applications with criteria such as families used in the same decision context [2,3,41,51].

According to data from the IBGE [49], the State of Mato Grosso do Sul is geographically located in the Midwest region of Brazil. There are four geographic meso-regions (Centro-Norte, Least, Southwest, and Pantanal) in this study's focus of analysis. The state has a privileged location and economic importance recognized due mainly to its diversity of crops such as cotton, rice, sugarcane, beans, cassava, wheat, corn, and soybean, with a predominance of the latter two, as well as its great biodiversity. These grains can be a source of bioeconomy and biofuel for the region, promoting sustainable development [52,53].

Sustainable development in the Pantanal region involves important environmental characteristics that must be considered when deciding the distribution of electricity. First, as suggested by Bergier et al. [21], the conservation of the Pantanal biome with new technologies, including electrical expansion, should be maximized. The Pantanal is the largest flooded area globally, located mostly in the state of Mato Grosso do Sul, where approximately 1,100,000 people live [23]. For this, the creation of actions and initiatives sought to equate the supply of complex isolated systems in the Pantanal of Mato Grosso do Sul, which presents difficulties of energy transmission via networks due to the large flooded zones and isolated points, as well as sugar cane, soybean, sugar, and cattle raising regions. In this way, solutions about the energy sector may contribute to the energy universalization for meso-regions of Mato Grosso do Sul and to the sustainable development of Brazil.

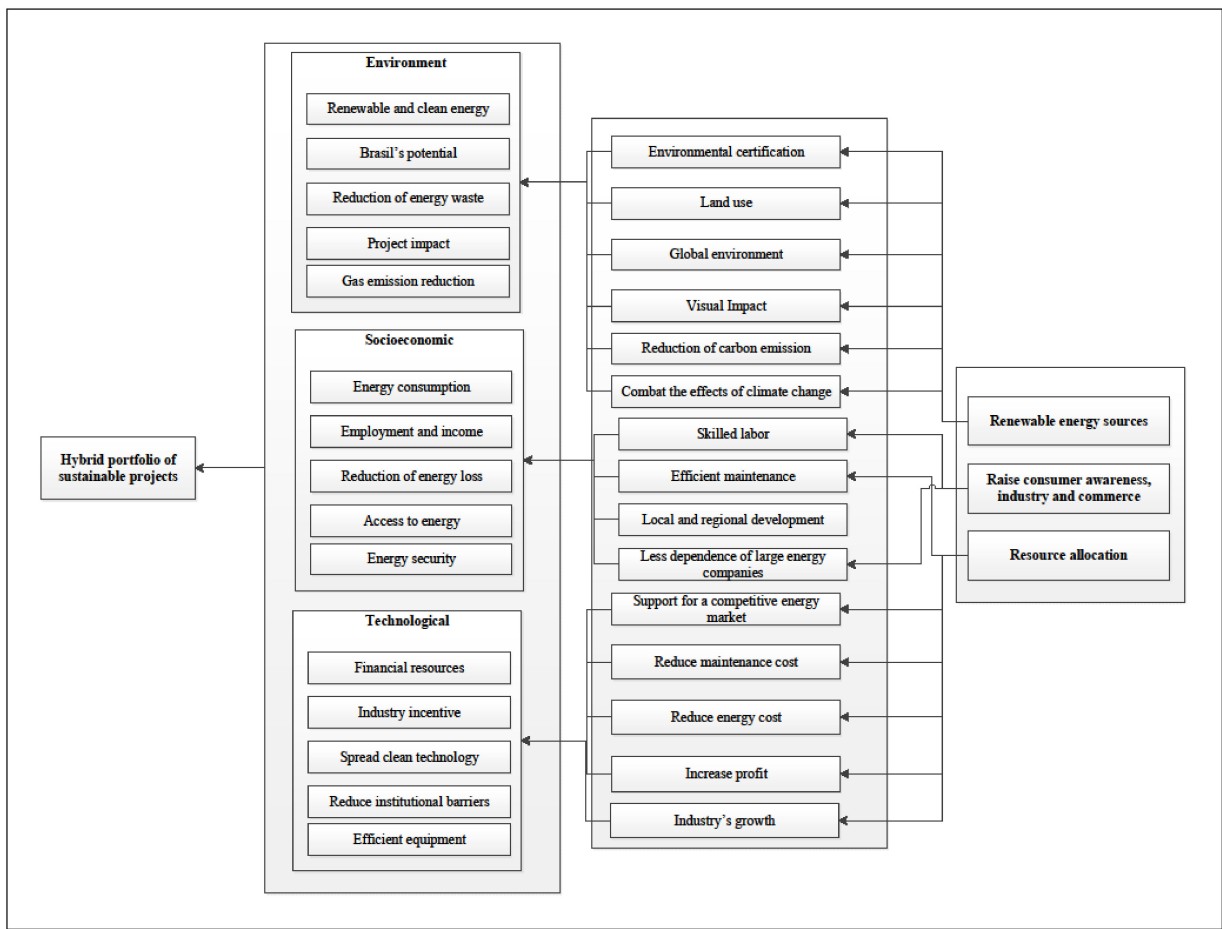

**Figure 2.** Structure and network of middle-end objectives linked to the core objectives.

In this first stage, 18 potential DGE projects were identified in selected locations, which have their space delimited in regional identity and the specificities organized around agriculture and cattle-raising, exploring the various possibilities of generation and supply. The following potential projects were identified: six photovoltaic solar energy distribution projects, six wind energy distribution projects, and six biomass energy distribution projects, valuing, therefore, the robustness of geographic dispersion of the potential projects for the State of Mato Grosso do Sul.

Figure 3 illustrates the geographic distribution of these projects and their installed capacities and simulated investment costs based on government data.

We have defined the action space, that is, the set of potential DGE projects in several selected localities, consisting of 18 potential DGE projects that include only renewable sources: wind, solar, and biomass. Each project has specificities relating to meso-regions of

Mato Grosso do Sul, as presented in Figure 3. Also, the set of projects to be considered in this study, in the framework for the second stage, consists of applying the multi-criteria decision model (MCDM/A), whose objective is to produce an appropriate recommendation to the decision-maker.

**Table 3.** Set of evaluation and description criteria.

| Criterion | Denotation | Description | Unity | Type |
|---|---|---|---|---|
| Required Area | C1 | It corresponds to the amount of area necessary to implement the energy generation source | $m^2$/kW | min |
| Greenhouse gases emission | C2 | It considers the amount of $CO_2$ emitted from energy generation sources | gCO2eq/kW | min |
| Contribution to the economic and regional development | C3 | It corresponds to the social and economic effects associated with the initiatives, such as the creation of new jobs, new companies in the supply chain, emerging companies in the energy sector, etc. | Qualitative (1–5) | max |
| Operation and maintenance costs | C4 | It considers the relation between the investment cost of the energy generation system and thegenerated energy | R$/kW | min |
| Payback | C5 | It is the investment payback time | Year | min |
| Technology efficiency | C6 | It corresponds to the amount percentage of electricity generated from the primary source of energy generation | % | max |
| Employment | C7 | It refers to the employment amount to be generated with the implantation of the energy generation project | Employment/Year/kW | max |

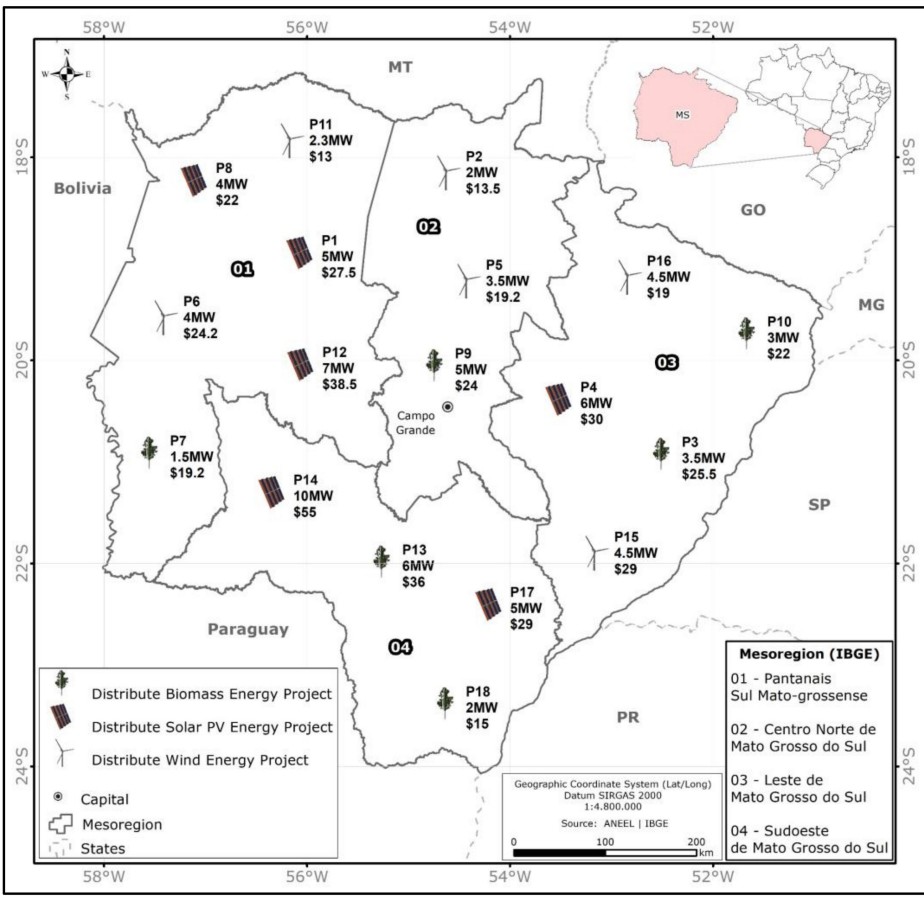

**Figure 3.** Geographical distribution of the DGE projects in MS.

The result of this stage contributes significantly to defining the volume of resources invested in the selected projects, always considering the restrictions of budgetary resources. Each project was evaluated according to seven criteria (see Table 3) and resulted in Figure 4.

The values of the consequences shown in Figure 4 were based on secondary governmental data. The weights of each criterion were considered equal since the criteria have equal relative importance for the decision in question.

| Scenario1 | C1 | C2 | C3 | C4 | C5 | C6 | C7 |
|---|---|---|---|---|---|---|---|
| Unit | (m²/kW) | (g CO2eq/kW) | (1-5) | (R$/kW) | (ano) | % | (emp./ano/kW) |
| Cluster/Group | ◆ | ◆ | ◆ | ◆ | ◆ | ◆ | ◆ |
| Preferences | | | | | | | |
| Statistics | | | | | | | |
| Evaluations | | | | | | | |
| P1 | 300,00 | 75,00 | 3,00 | 0,33 | 12,00 | 60,00 | 1,20 |
| P2 | 480,00 | 35,00 | 4,00 | 0,15 | 9,00 | 68,00 | 1,60 |
| P3 | 4306,00 | 100,00 | 4,00 | 0,30 | 12,00 | 75,00 | 2,00 |
| P4 | 290,00 | 40,00 | 5,00 | 0,34 | 8,00 | 60,00 | 1,20 |
| P5 | 680,00 | 40,00 | 5,00 | 0,25 | 9,00 | 68,00 | 1,60 |
| P6 | 230,00 | 35,00 | 3,00 | 0,49 | 11,00 | 68,00 | 1,60 |
| P7 | 4548,00 | 125,00 | 4,00 | 0,24 | 8,00 | 75,00 | 2,00 |
| P8 | 320,00 | 80,00 | 3,00 | 0,23 | 8,00 | 75,00 | 1,20 |
| P9 | 4000,00 | 120,00 | 3,00 | 0,26 | 12,00 | 68,00 | 2,00 |
| P10 | 3187,00 | 100,00 | 3,00 | 0,23 | 9,00 | 60,00 | 2,00 |
| P11 | 290,00 | 45,00 | 4,00 | 0,16 | 8,00 | 75,00 | 1,60 |
| P12 | 200,00 | 95,00 | 5,00 | 0,64 | 9,00 | 60,00 | 1,20 |
| P13 | 4210,00 | 100,00 | 3,00 | 0,47 | 12,00 | 68,00 | 2,00 |
| P14 | 490,00 | 60,00 | 4,00 | 0,57 | 9,00 | 60,00 | 1,20 |
| P15 | 980,00 | 35,00 | 4,00 | 0,34 | 11,00 | 68,00 | 1,60 |
| P16 | 780,00 | 35,00 | 4,00 | 0,21 | 10,00 | 68,00 | 1,60 |
| P17 | 320,00 | 60,00 | 5,00 | 0,31 | 9,00 | 60,00 | 1,20 |
| P18 | 4283,00 | 120,00 | 4,00 | 0,19 | 8,00 | 75,00 | 2,00 |

**Figure 4.** Consequence matrix of the eighteen projects evaluated by the criteria.

As a preliminary result, in the second stage of the framework, a projects order was obtained through the PROMETHEE II technique (Figure 5), establishing a recommendation for the decision-maker for those projects that should be prioritized according to their preferences and values.

Based on the analysis of the ranked projects, the last stage of the framework consists of performing an analysis of the investment cost of these sustainable projects, as shown in Figure 3, considering the available financial resources, which are restricted. The sustainable projects obtained in this study are evaluated to define which can be implemented according to the budget constraint that corresponds to 40% of the total investment cost in DGE projects earmarked for sustainability (i.e., R$916.00); the PROMETHEE V technique was used.

From the net flows obtained in Stage 2 through PROMETHEE II (Figure 1), it was possible to formulate an integer linear programming problem (ILPP), together with the problem budget constraint. As a result of the PROMETHEE V technique, a portfolio with nine projects was obtained {P2, P4, P5, P8, P11, P16, P18}, with the first column indicating the projects, the second column indicating the net flow of each project, and the third column reading "yes" for selected projects and "no" for non-selected projects, according to Figure 6, detailing two {P4, P8} photovoltaic projects, four {P2, P5, P11, P16} wind projects, and two {P7, P18} biomass projects.

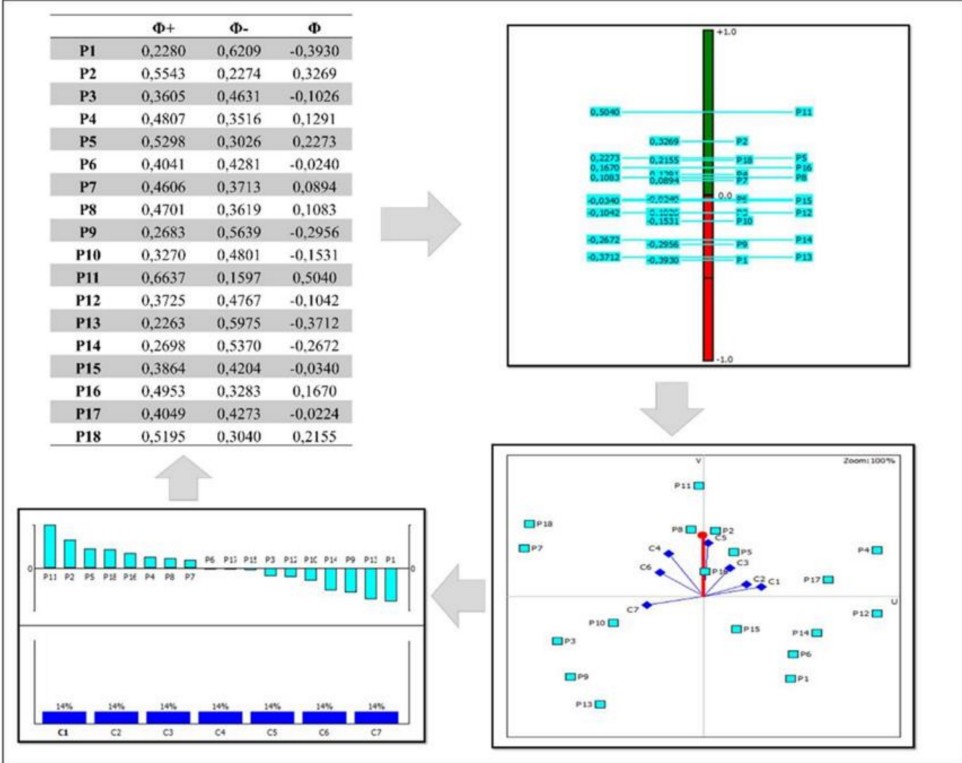

**Figure 5.** Result PROMETHEE II for set of the DGE projects by the set of criteria used.

| Actions | | Net Flow | Optimal | Compare |
|---------|---|----------|---------|---------|
| | | **Total:** | **1,7731** | **1,7731** |
| P1 | | -0,3950 | no | no |
| P2 | | 0,3277 | yes | yes |
| P3 | | -0,1008 | no | no |
| P4 | | 0,1261 | yes | yes |
| P5 | | 0,2269 | yes | yes |
| P6 | | -0,0252 | no | no |
| P7 | | 0,0924 | yes | yes |
| P8 | | 0,1092 | yes | yes |
| P9 | | -0,2941 | no | no |
| P10 | | -0,1513 | no | no |
| P11 | | 0,5042 | yes | yes |
| P12 | | -0,1092 | no | no |
| P13 | | -0,3697 | no | no |
| P14 | | -0,2689 | no | no |
| P15 | | -0,0336 | no | no |
| P16 | | 0,1681 | yes | yes |
| P17 | | -0,0252 | no | no |
| P18 | | 0,2185 | yes | yes |

**Figure 6.** Result PROMETHEE V: selected DGE projects.

The optimization procedure step supported by PROMETHEE V provides the solution to the problem, presenting a portfolio composed of a subset of projects that maximizes its value at a score of 1.77 and meets the budget constraints of the problem with a value of R$744.00, lower than the total budget of R$916.00, to the decision-maker. In the case applied, a *V(pr)* = {P2, P4, P5, P8, P11, P16, P18} portfolio was given to the decision-maker.

## 5. Discussion and Managerial Implications

With the combination of VFT and PROMETHEE techniques, it was possible to rank the best energy generation options for each meso-region, emphasizing the spatial and geographic dispersion of the State of Mato Grosso do Sul. Thus, the validation of the proposed model verifies whether the best generation options are available for distributed energy systems to meet regional characteristics. For this, in the next paragraphs, the agreement of the best options of distributed energy generation projects with the characteristics of each meso-region are verified.

In this way, the projects P2 and P5 are wind projects (in the vicinity of the cities of Sonora and Coxim) located in the northern central region of the State of Mato Grosso do Sul. Its economic base is centered on agriculture, with a predominance of large rural properties, and with emphasis appearing in the municipalities of São Gabriel do Oeste, Costa Rica, and Sonora, which, together, account for 88.8% of grain production in the region, with higher expressions being soybean, corn, and cotton crops. Thus, the result can be explained because the selection of wind sources in the region avoids the use of water from dams that generate electricity in the region, considering that in recent years these reservoirs have been decreasing [52]. Thus, such grain crops, requiring irrigation at certain times of the year, can respect the flows of the region's rivers and, at the same time, have electricity from renewable technologies, respecting other regional characteristics [54].

In the East region, the P4 (solar) and P16 (wind) projects were the best options observed in the results. This is a region that developed within agropastoral characteristics with a strong predominance of livestock. Agriculture in the region, except for the municipality of Chapadão do Sul, is low dynamic and has always played a secondary role in the economy of most of its municipalities. Thus, two aspects can contribute to project choices: the promotion of jobs from renewable sources and the use of extensive livestock spaces. Thus, for cities with small-scale farmers' economies, solar and wind project facilities can be an additional source of income and value creation for local production [55,56].

For the Pantanal region, the projects recommended from the model results were P8 and P11, both wind projects. The regional economic occupation is mainly centered on the extensive cultivation of beef cattle and agriculture, which is, due to soil characteristics, only the production of irrigated rice in the municipality of Miranda. Thus, wind projects can respect more than 80% of the preserved biome of the region, as they will be sources of income and development of a region, in which it has a large plain area and great biodiversity, requiring renewable energy alternatives to meet local demand [22,56,57].

Finally, the P18 Biomass project was selected as the best option for the Southwest meso-region of the State of Mato Grosso do Sul (MS) near the municipality of Sete Quedas. The Southwest Meso-region has a strong economic dependence on rural production. Cassava is of local importance for feeding the population and in producing raw material for the food industry in the region, and its residue is an alternative potential for generating energy from biomass. Furthermore, beef cattle are an important activity in the occupation of areas with little agricultural aptitude, which highlights the development of biomass projects for electricity generation. The municipality of Sete Quedas has a characteristic focused on agribusiness, its activities in cattle ranching. Thus, the selection of distributed energy generation projects based on biomass can enhance the region's economy, integrating agricultural and livestock production without the need for more land to increase the region's productivity [58].

Brazil is one of the largest producers of sugarcane and meat globally, and, thus, has the potential to simultaneously reduce its emissions, use the land more efficiently, and

exchange large quantities of fossil fuels. Thus, the choice for biomass for this region becomes justifiable from the VFT-PROMETHEE techniques applied. Furthermore, this meso-region can become a major supplier of biofuel to other sectors of the Brazilian economy [59], in addition to reducing the impacts on greenhouse gas emissions caused by livestock farming [59].

Thus, the Framework proposed in Figure 1 based on Value-Focused Thinking (VFT) and PROMETHEE V techniques can be considered a valid model for sustainable decisions in the energy sector. Its characteristics allow for considering sustainability in the alternative sources portfolio in the energy sector. This can be explained by the fact that the decisions and results observed are more clearly defined and the decisions become more efficient when assessing the problem [60,61].

There is a range of studies in the literature with emphasis on clean and renewable energy, as well as tool guides and methodologies for project portfolio selection in the energy sector [2,3,14,26–32]. However, the contribution of this study is in the hybrid combination of VFT and MCDM/A methodology applied to create the projects with an emphasis on the geographical localization of Mato Grosso do Sul (Brazil) and for the evaluation and selection of these defined projects to be implemented. These are analyzed to ensure that they meet the combination of objectives (values) defined by the decision-maker and meet the geographic needs of the Mato Grosso do Sul region. Otherwise, one only considers the implementation of wind, solar, and biomass projects and imposes it on the decision-maker without evaluating its values and the geographic location to be implemented. In this sense, the VFT methodology would be applied, this being the study's objective: the use of VFT methodology applied in the portfolio selection of DGE projects with multi-criteria evaluation.

In summary, the application of the proposed model can be recommended. Managers and decision-makers can apply the multi-criteria model to improve the choice of distributed energy generation projects in other parts of the country. For this, managers can initially outline the alternatives for each meso-region studied, that is, which technologies will meet the region's energy needs. From the technologies chosen in the portfolio composition, managers can gather information from stakeholders and define the main criteria that will be evaluated and analyzed in each project. Finally, it is necessary to understand that the projects chosen from the model satisfy local needs and promote sustainable development.

The multi-criteria model was applied later to rank the projects and then select a subset of projects against the budget constraints, thus recommending those that can be implemented. The decision-making process in the management of the energy sector, therefore, has gained another contour yet: the new technologies associated with the process of energy generation, transmission, and distribution,. Such characteristics naturally lead this type of generation to the small size, exploring solar and wind sources, which composes the renewable energy panel.

As indicated by Aquila et al. [43], the energy sector is more complex, making it critical to reduce inventories through new clean energy generation technologies projects. Therefore, the VFT methodology associated with the MCDM/A model, in this study PROMETHEE, when applied to the energy sector, enables contemplation of the pillars of sustainability (economic, environmental, and social) and improving the decision-making process of energy sector managers.

## 6. Conclusions

This paper aims to propose a multi-criteria model to support decision-making from a portfolio in selecting technologies for DGE projects based on the characteristics of the geographic space in Brazil. The MCDM/A approach combined with VFT was used, being the PROMETHEE II and V techniques which were applied to conduct this evaluation, as they can provide an evaluation of alternatives based on the decision-maker's preference elicitation, considering the trade-offs among the criteria and the selection of the set of projects to be implemented initially given the budget constraint.

The combination of VFT and PROMETHEE techniques allowed us to evaluate the technological alternatives for energy generation satisfactorily according to the meso-regions of the State of Mato Grosso do Sul. Thus, the results of this article suggest that the combination of techniques allows managers to capture the stakeholder values and, above all, take account of local characteristics where the projects will be executed. Thus, managers will be able to use the combination of techniques to also promote sustainable development.

The results of this paper may influence studies to support and clarify energy companies that invest in DGE projects around the world and national development, mainly in the identification of opportunities for decision-making and the geographical location for investment in the generation of alternative of renewable energy sources. Moreover, another point consists of considering multiple criteria in the economic, social, environmental, and technological dimensions for the evaluation and selection of these projects, aligning with the Sustainable Development Goals (SDG).

In this perspective, it is observed that the multi-criteria decision techniques are relevant tools that may be applied to support decision-makers in the decision process in the context of sustainability in the electric energy distributed generation. It is worth mentioning that the application made in this paper considered only a need to validate the model with realistic data from the energy sector. Therefore, the application used can be considered a limitation of the study.

**Author Contributions:** M.B.: conceptualization, methodology, formal analysis, software, writing—original draft preparation, writing—review and editing. M.F.: writing— software, original draft preparation, visualization, writing—review and editing. S.G.C.: visualization and supervision. T.M.A.: data collection and investigation. C.C.d.S.: visualization and supervision. J.F.d.R.N.: visualization and supervision. J.F.d.F.: visualization and supervision. All authors have read and agreed to the published version of the manuscript.

**Funding:** This research and APC were funded by Edict number 26/2021-Pró-Reitora de Pesquisa e Pós-Graduação (PROPP)/Universidade Federal de Mato Grosso do Sul (UFMS), protocolo R6SCX.290921.

**Institutional Review Board Statement:** Not applicable.

**Informed Consent Statement:** Not applicable.

**Data Availability Statement:** Data available on request due to privacy/ethical restrictions.

**Acknowledgments:** The authors would like to acknowledge the Coordenação de Aperfeiçoamento de Pessoal de Nível Superior-Brasil (CAPES)-Finance Code 001, and Universidade Federal de Mato Grosso do Sul (UFMS) and Anhanguera University-Uniderp.

**Conflicts of Interest:** The authors declare no conflict of interest. The funders had no role in the design of the study; in the collection, or in the decision to publish the results.

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
