# Peer review of "Combining Value-Focused Thinking and PROMETHEE Techniques for Selecting a Portfolio of Distributed Energy Generation Projects in the Brazilian Electricity Sector"

_sustainability, doi:10.3390/su131911091_

Round 1
Reviewer 1 Report
The article is technical, it appears interesting both from a theoretical point of view and for its practical uses.
The inclusion of a short paragraph in which the case study is also better explained, also geographically, is suggested. Some explanatory maps and images could be added. The case study is briefly described only in few lines (370-374)
In lines 71-73 be more explicit about these specifica features of the case study.
Also some more detailed socio-economic data could improve the description of the case study.
The matherials and methods are very well described and the figure helps to understand the process. However, the paragraph 3.4 could be more explicit about the methods used to analyse and define the stakeholders.
Concerning the discussion, would it be possible to define more specific guidelines or recommendations at the regional level for policy makers and stakeholders?
Please check the style of the bibliography.
Author Response
To the Reviewer #1
We thank you for the comments and the suggestions for improvement of our paper. We have made all the suggestions that the reviewer proposed to us.
1.1) Reviewer’s Comments: “The article is technical, it appears interesting both from a theoretical point of view and for its practical uses.”
Authors’ response: Thanks for the comment and for the potential of the paper. We hope to publish the paper soon to share it with the scientific community.
1.2) Reviewer’s Comments: “The inclusion of a short paragraph in which the case study is also better explained, also geographically, is suggested. Some explanatory maps and images could be added. The case study is briefly described only in few lines (370-374)” and “In lines 71-73 be more explicit about these specifica features of the case study.” and “Also some more detailed socio-economic data could improve the description of the case study.”
Authors’ response: We agree with the reviewer and follow his recommendation. We have inserted one paragraphs.
- Section 4.1 (page 9, 329 – 344 lines): “The state of Mato Grosso do Sul is the 6th state in the country in territorial extension, with 357,145,534 km2 which corresponds to 4.19% of the total area of Brazil (8,515,767.049 km2 ) and 22.23% of the area of the central region -West. Its estimated population in 2021 is 2,839,188 inhabitants, making the state the 21st population in Brazil. Its capital and largest city is Campo Grande, and other important municipalities are Dourados, Três Lagoas, Corumbá, Ponta Porã, Aquidauana, Nova Andradina and Naviraí. The intrinsic character of the micro and mesoregional divisions of Mato Grosso do Sul refer to a set of economic, social and political determinations that concern the entire organization of space in the state territory, with the aim of assisting in the development of public policies, of planning, subsidize regionalized and local studies (IBGE, 2021). In the energy sector, in 2018, the state's electricity consumption and consumers were 5,788,274 MWh and 1,089,139 consumers, respectively. (SEMAGRO, 2019). The State of Mato Grosso do Sul has 2 Hydroelectric Power Plants, 9 Small Hydroelectric Power Plants, 1 solar photovoltaic generating plant, 21 thermoelectric power plants based on natural gas in the fossil class and 28 thermoelectric power plants using sugarcane bagasse as fuel in the Biomass class.”
1.3) Reviewer’s Comments: “The matherials and methods are very well described and the figure helps to understand the process. However, the paragraph 3.4 could be more explicit about the methods used to analyse and define the stakeholders.”
Authors’ response: We have inserted one paragraph.
- Section 3.4 (page 7, 282 – 289 lines): “The definition of the actors in the decision-making process follows the methodology proposed by De Almeida et al. (2016). Regarding the definition of the decision-making and stakeholders, the energy project manager was assigned the role of decision-makers, while the operational leaders of each of the sectors, as well as the analyst of methods and processes were characterized as experts. In addition, it is important to evaluate the presence of stakeholders (such as shareholders and government representatives) and whose influence must be carefully managed in order to prioritize the interests of the organization.”
1.4) Reviewer’s Comments: “Concerning the discussion, would it be possible to define more specific guidelines or recommendations at the regional level for policy makers and stakeholders?”
Authors’ response: We agree with the reviewer and follow his recommendation.
- Section 5 (pages 17-18, 539 – 550 lines): “In summary, the application of the proposed model can be recommended. Managers and decision makers can apply the multi-criteria model to improve the choice of distributed energy generation projects in other parts of the country. For this, managers can initially outline what are the alternatives for each mesoregion studied, that is, which technologies will meet the region's energy needs. From the technologies chosen in the composition of the portfolio, managers can gather information from stakeholders and define the main criteria that will be evaluated and analyzed in each project. Finally, it is necessary to understand that the projects chosen from the model satisfy local needs and promote sustainable development.”
1.5) Reviewer’s Comments: “Please check the style of the bibliography.”
Authors’ response: We appreciate the reviewer's careful look.

Reviewer 2 Report
The authors address an interesting and up-to-date topic developing a decision-making framework based o Value-Focused Thinking and PROMETHEE methods to select the best electric energy distributed generation project to invest in. The development of such a decision-making framework is interesting and engaging. Some of the paragraphs or parts of the manuscript need to be explained more clearly. Although the topic is interesting, several major shortcomings need to be improved.
Suggestions for improvement:
- The manuscript should be set according to the Journal’s template and instruction to authors (text, figures, tables, equations, references, abbreviations, etc.). Please define abbreviations in the text before their first use, variables, SI units, figures, etc.
- Consistently use terminology throughout the manuscript (particular examples: distributed energy generation projects or distributed generation on electricity projects; PROMETHEE techniques or PROMETHEE methods)
- Check and improve the English language and grammar throughout the paper (check misspellings, writing in the first person, etc.), as well as all figures and tables (both must be readable). The authors should be consistent in writing
- The introduction provides sufficient background information but lacks clearly stated research goals and hypotheses. Try to include additional relevant references. Some fundamental references are missing as well as the recent ones considering the research problem. For example, there is no mention of Jajac, Marović, Macharis, or Waaub who published on the MCDA, particularly PROMETHEE, in the multi-stakeholder project environment. The research problem is not clear while the research goals and hypotheses are not clearly stated
- The literature review should be improved. At the moment it lacks a critical overview of the other approaches in solving the stated research problem and the methodology upgrade that is proposed by this research
- The research design is not clearly written. The research methodology should be clear and the hows and the whys of used methods should be clearly visible. The validation is especially important, therefore authors are urged to give insight into the validation process. Also, add some additional discussion of findings in relation to the research framework as well as research goals and hypotheses are needed
- The results and discussion section lack clarity. It is well balanced and written but some additional emphasis should be given in relation to newly added and recent references as well as to elaborate all used abbreviations. Such should be clearly connected to previously stated goals as well further elaborated in the discussion. The major problem of the proposed approach seems to be its validation
- The authors are encouraged to draw more specific conclusions in relation to the research framework as well as research goals and hypotheses. Right now it looks like good observations that are not aligned with the research goals
Overall, I believe that the article provides valuable content to the present body of knowledge but should be written in a more clear, consistent, and compelling way. Therefore at the moment, the manuscript does not reach the desired level for publishing. I strongly urge the authors to reconsider the above-mentioned comments, rewrite the paper accordingly, and resubmit.
Author Response
Reviewers' comments
To the Reviewer #2
We thank you for the comments and the suggestions for improvement of our paper. We have made all the suggestions that the reviewer proposed to us.
2.1) Reviewer’s Comments: “The authors address an interesting and up-to-date topic developing a decision-making framework based o Value-Focused Thinking and PROMETHEE methods to select the best electric energy distributed generation project to invest in. The development of such a decision-making framework is interesting and engaging. Some of the paragraphs or parts of the manuscript need to be explained more clearly. Although the topic is interesting, several major shortcomings need to be improved.”
Authors’ response: We welcome the reviewer's comments on the relevance of the study. This keeps us engaged in research that contributes to the advancement of the study area. The adjustments made in the article are explained in the next comments.
2.2) Reviewer’s Comments: “Consistently use terminology throughout the manuscript (particular examples: distributed energy generation projects or distributed generation on electricity projects; PROMETHEE techniques or PROMETHEE methods).”
Authors’ response: Thank you for this suggestion. We made a requested conform subscription requested, maintaining a pattern in the use of terminologies.
2.3) Reviewer’s Comments: “Check and improve the English language and grammar throughout the paper (check misspellings, writing in the first person, etc.), as well as all figures and tables (both must be readable). The authors should be consistent in writing.”
Authors’ response: We have fully reviewed the article. The analysis was made from the hiring of a native English service made available by the publisher itself.
2.4) Reviewer’s Comments: “he introduction provides sufficient background information but lacks clearly stated research goals and hypotheses. Try to include additional relevant references. Some fundamental references are missing as well as the recent ones considering the research problem. For example, there is no mention of Jajac, Marović, Macharis, or Waaub who published on the MCDA, particularly PROMETHEE, in the multi-stakeholder project environment. The research problem is not clear while the research goals and hypotheses are not clearly stated.”
Authors’ response: We welcome the reviewer's comments on the relevance of the study.
Section 1 (page 2): We clarify the goal of the research, as well as use arguments based on relevant references such as: Jajac, N., Marović, I., & Mladineo, M. (2014). Planning support concept to implementation of sustainable parking development projects in ancient Mediterranean cities. Croatian Operational Research Review, 5(2), 345–359. doi:10.17535/crorr.2014.0018. These authors allowed us to clarify the state of the art of this study, as well as research goals.
- Section 1 (page 2, 59 – 67): “However, there is no known application in the domain of portfolio selection of a DGE projects environmental, economic, and social constraints in developing geographical regions. Study the complement of hydroelectric plants, selecting DGE projects and identifying their role in the electro-energy dispatch of the State of Mato Grosso do Sul, as well as how different regions of the state as potential generation of clean and accessible energy for the most remote regions of the state, this being a planning problem, which makes it a problem of multicriteria decision, which involves a large number of alternatives, multiple criteria, as well as stakeholders of the decision-making process, as well as application of the mcdm/a conform technique explained by Jajac, Marović, Macharis (2014)”.
- Section 1 (page 2, 73-78 lines): “To achieve this goal, the proposed model is based on the combination of VFT and PROMETHEE techniques whose application allows to take in sustainability characteristics where the best options for power generation will be chosen. This paper advances in knowledge because consider the structuring in the decision-making process of feasible alternatives generation for portfolio selection in the sustainable electric energy distributed generation context.”
2.5) Reviewer’s Comments: “The literature review should be improved. At the moment it lacks a critical overview of the other approaches in solving the stated research problem and the methodology upgrade that is proposed by this research.”
Authors’ response: We agree with the reviewer and follow his recommendation. We improved the literature review and made a link to the literature and research problem in which we provided a basis for the proposition of the methodology. Section 2 (page 3-4, 121– 204 lines.
2.6) Reviewer’s Comments: “The research design is not clearly written. The research methodology should be clear and the hows and the whys of used methods should be clearly visible. The validation is especially important, therefore authors are urged to give insight into the validation process. Also, add some additional discussion of findings in relation to the research framework as well as research goals and hypotheses are needed.”
Authors’ response: We agree with the reviewer and follow his recommendation. We seek to describe based on relevant references section 2, as advantages of the methods used, as well as the validation process of the table in section 5.
- Section 5 (pages 17-18, 539 – 550 lines): “In summary, the application of the proposed model can be recommended. Managers and decision makers can apply the multi-criteria model to improve the choice of distributed energy generation projects in other parts of the country. For this, managers can initially outline what are the alternatives for each mesoregion studied, that is, which technologies will meet the region's energy needs. From the technologies chosen in the composition of the portfolio, managers can gather information from stakeholders and define the main criteria that will be evaluated and analyzed in each project. Finally, it is necessary to understand that the projects chosen from the model satisfy local needs and promote sustainable development.”
2.7) Reviewer’s Comments: “The results and discussion section lack clarity. It is well balanced and written but some additional emphasis should be given in relation to newly added and recent references as well as to elaborate all used abbreviations. Such should be clearly connected to previously stated goals as well further elaborated in the discussion. The major problem of the proposed approach seems to be its validation.”
Authors’ response: Thank you for this suggestion. We agree with the reviewer and follow his recommendation.
- Sectoin 5 (page 16, 474 – 480 lines): We explain that the proposed model is validated based on the discussion about the agreement between the best alternatives and the regional characteristics of each mesoregion.
The excerpt inserted in the Discussion section was: “With the combination of VFT and PROMETHEE techniques, it was possible to rank the best energy generation options for each mesoregion with emphasis on the spatial and geographic dispersion of the State of Mato Grosso do Sul. Thus, the validation of the proposed model verifies whether the best generation options are available distributed energy systems meet regional characteristics. For this, in the next paragraphs, it is verified the agreement of the best options of distributed energy generation projects with the characteristics of each mesoregion.”
2.8) Reviewer’s Comments: “The authors are encouraged to draw more specific conclusions in relation to the research framework as well as research goals and hypotheses. Right now it looks like good observations that are not aligned with the research goals.”
Authors’ response: Thank you for this suggestion. We agree with the reviewer and follow his recommendation.
- Section 6 (pages 18, 582 - 588 lines): We insert one more paragraph in the conclusion to align research objective, study techniques and observed results.
The excerpt inserted in the Conclusion section was: “The combination of VFT and PROMETHEE techniques allowed to satisfactorily evaluate the technological alternatives for energy generation according to the mesoregions of the State of Mato Grosso do Sul. Thus, the results of this article suggest that the combination of techniques allows managers to capture the stakeholder values and, above all, take into account the local characteristics where the projects will be executed. Thus, managers will be able to use the combination of techniques to also promote sustainable development.”
2.9) Reviewer’s Comments: “Overall, I believe that the article provides valuable content to the present body of knowledge but should be written in a more clear, consistent, and compelling way. Therefore at the moment, the manuscript does not reach the desired level for publishing. I strongly urge the authors to reconsider the above-mentioned comments, rewrite the paper accordingly, and resubmit.”
Authors’ response: We hope that the revisions made have met all the suggestions making the article clearer and more objective. We appreciate all the comments made by Reviewer #2 and the acknowledgment of the relevance of the study.

Round 2
Reviewer 2 Report
In the revised version authors gave additional insights into their research and also acted upon given comments and suggestions, and gave all required clarifications. Overall, I believe that the article provides valuable content to the present body-of-knowledge.